# Synthetic Colors in Food: A Warning for Children’s Health

**DOI:** 10.3390/ijerph21060682

**Published:** 2024-05-27

**Authors:** Zandleme Birino de Oliveira, Darlene Vitória Silva da Costa, Ana Caroline da Silva dos Santos, Antônio Quaresma da Silva Júnior, Amanda de Lima Silva, Raphael Carlos Ferrer de Santana, Isabella Cristhina Gonçalves Costa, Sara Freitas de Sousa Ramos, Gabriel Padilla, Silvia Katrine Rabelo da Silva

**Affiliations:** 1Programa de Pós-Graduação em Recursos Naturais da Amazônia, Universidade Federal do Oeste do Pará, Santarém 68040-255, PA, Brazil; katrinerabelos@gmail.com; 2Laboratório de Microbiologia, Instituto de Saúde Coletiva, Universidade Federal do Oeste do Pará, Santarém 68040-255, PA, Brazil; darlenecosta.biologa@gmail.com (D.V.S.d.C.); dossantosanacarolinedasilva@gmail.com (A.C.d.S.d.S.); delimasamanda@gmail.com (A.d.L.S.); raphaell.ferrer@gmail.com (R.C.F.d.S.); isabellacgcosta@gmail.com (I.C.G.C.); sara_freitas_stm@hotmail.com (S.F.d.S.R.); 3Programa de Pós-Graduação em Biodiversidade e Biotecnologia da Rede Bionorte, Universidade Federal do Oeste do Pará, Santarém 68040-255, PA, Brazil; antoniojuniort6@hotmail.com; 4Departamento de Microbiologia, Instituto de Ciências Biomédicas, Universidade de São Paulo, São Paulo 05508-000, SP, Brazil; gabrielpadilla@usp.br; 5Programa de Pós-Graduação em Ciências da Saúde, Universidade Federal do Oeste do Pará, Santarém 68040-255, PA, Brazil

**Keywords:** synthetic dyes, food additives, toxicity, behavioral disorders

## Abstract

This study addressed the harmful effects of artificial colors in pediatric populations, including children diagnosed with Autism Spectrum Disorder (ASD) and Attention Deficit Hyperactivity Disorder (ADHD), as well as those without behavioral disorders. There is a consensus that synthetic food colorings have several impacts on consumers, especially pediatrics, due to their influence on sensory appeal, which can encourage preference for certain foods. The results revealed that these color additives are directly linked to a series of health problems, with a greater impact on children, including a predisposition to pathological conditions such as carcinogenic, allergenic, mutagenic, cytotoxic, and clastogenic activities, as well as gastrointestinal and respiratory problems, in addition to behavioral changes in children with and without diagnosed disorders. The harms of synthetic dyes in children with or without comorbidities are worrying and require a careful and proactive approach from parents, caregivers and public authorities.

## 1. Introduction

The coloring present in food has several impacts on consumers. A more vibrant color sharpens the sensorial appeal and encourages preference for a certain food, as this attribute is directly linked to the excellence of the product, its vitality and even its nutritional composition [1]. However, the artificial colors used for these purposes are designed for the unique purpose of boosting sales, and they are devoid of any nutritional benefit.

Many studies indicate that artificial colors are directly associated with various pathological conditions, such as cancer, food allergies, and gastrointestinal and respiratory problems. Furthermore, behavioral changes in children are described, with the group of children with Autism Spectrum Disorder (ASD) being one of the most impacted by these adversities [2,3].

Autism is a neurodevelopmental disorder that affects communication and social behavior and causes restricted and repetitive interest patterns in terms of eating and activities [3]. It is characterized by difficulties in social interaction, as well as difficulties in verbal and non-verbal communication. Autism is considered a complex disorder, influenced by genetic, environmental and neurobiological factors [4].

A child’s neurodevelopment has a direct influence on several behavioral aspects throughout their life, covering their social interaction, language, hearing, vision, behavior and posture. Essentially, the proper functioning of this development is crucial to ensuring satisfactory performance in the school environment. When neurodevelopment occurs regularly, the child is expected to achieve significant academic performance. However, if there are irregularities in this process, it is likely that the child will need more attention to follow their peers in academic activities [5].

The group of people with Autism Spectrum Disorder is more vulnerable to the negative effects of food additives due to the behavioral pattern known as Avoidant/Restrictive Food Intake Disorder (ARFID). This disorder manifests through a range of eating behaviors, including overeating, severe restrictions and even the complete avoidance of certain foods by the affected person, especially children, since their interests, for the most part, are stereotyped and they tend to maintain repetitive eating patterns, often based on colors, accepting some and rejecting others [6,7].

Sensitivity to color additives is not exclusive to the ASD group. Children without behavioral disorders may also manifest neurobehavioral abnormalities when consuming synthetic dyes [8]. Therefore, children diagnosed with Attention Deficit Hyperactivity Disorder (ADHD) also experience changes in behavior. According to Rambler et al. [2], children exposed daily to synthetic dyes face significant consequences, presenting statistical changes in symptoms.

We present in this document an updated review based on research evidence about the harm of artificial colors in pediatric populations diagnosed with ASD and ADHD and in children without behavioral disorders.

## 2. Materials and Methods

This study consists of a comprehensive analysis of the literature, in which a search for bibliographic references was conducted, including articles published between 2020 and 2023, following the approach outlined by Braga and Melo [9]. The findings were debated among the work’s collaborators and unanimously chosen to be incorporated into this review. The evaluation considered criteria such as the methodological quality of the articles, taking into account the characteristics of the sample, the type of intervention and comparator, in addition to the follow-up time, level of evidence (LE) and strength of recommendation (SR). To achieve this objective, searches were carried out in electronically indexed databases, including the National Library of Medicine of the United States—National Institutes of Health (PubMed), Web of Science, Scopus, Lilacs and Medline.

To consult scientific articles and delve deeper into the topic, the following descriptors were used: Synthetic food coloring, Behavior, Processed foods, Food additives, Toxicity, Artificial food coloring, Attention deficit hyperactivity disorder, Behavioral problems in childhood, Autism spectrum disorder, Avoidant/restrictive food intake disorder, Behavioral disorders, Neurobehavioral abnormalities. The methodology for preparing this work can be seen in detail in Figure 1.

## 3. Results

After a thorough search for relevant articles, which was carried out in scientific databases, 83 studies were initially identified that could contribute to the review. After removing duplicates and excluding articles with a publication date greater than 5 years, 15 studies remained. The studies were selected for the review, as they were aligned with the objective of the work, focused on the harm of artificial dyes in pediatric populations diagnosed with Autism Spectrum Disorder (ASD) and Attention Deficit Hyperactivity Disorder (ADHD) and in children without behavioral disorders. The selected studies were organized in a format that allowed for a more accurate clinical view of the effects of artificial colors in pediatric populations. The results of these studies are summarized and presented in Table 1.

According to the literature data, the first synthetic dyes were developed in 1771 by the Irish chemist Peter Woulfe [1]. Since then, over the following years, there has been a significant increase in the variety of colors available for foods, resulting in a gradual transition from natural-origin dyes, such as plants, flowers and root extracts, to synthetic ones [10]. This broad spectrum of colors has had a direct impact on the food industry, providing a diverse range of colors to foods, which has increasingly attracted consumers. This phenomenon resulted in increasing profits, in addition to influencing customers’ perception, as they began to associate the new colors with the quality of the products [11]. Furthermore, a significant increase in the use of these additives has been observed, with estimates indicating that more than 8 million tons of more than 2000 types of synthetic dyes are used annually across the world [12].

Brazil is one of the world leaders in terms of the use of synthetic dyes, with a total of 11 types allowed by the Health Surveillance Agency (ANVISA) for daily use in the food industry. In comparison, the United States regulates only seven synthetic dyes released by the Food and Drug Administration (FDA), which means that Brazil regulates almost double the number of these additives. However, since the 1960s, research has showed that some of the dyes considered safe for consumption by the ANVISA already showed signs of toxicity after human consumption [1]. An example is Tartrazine, which has carcinogenic activity and genotoxic potential, and whose toxic effects had already been reported since the 1990s by Giri et al. [13] and in the 2000s by Mpountoukas et al. [14].

Several studies by different researchers and in different countries have associated problems related to mutagenic activities with the chronic use of several of the dyes consumed daily by the Brazilian population, including Sunset Yellow, Amaranth, Erythrosine and Indigotine Blue [15,16,17,18]. A single dye, Red 40, has been observed to be harmful to health in several respects, with side effects including kidney, stomach and lung diseases and anemia [19]. In addition to the mutagenic effects, other studies also reported allergenic activities associated with the consumption of these substances, especially in pediatric populations [20].

A warning has been issued about the use of synthetic dyes, especially aimed at parents and caregivers of children with ASD and ADHD. The basis for this concern lies in studies that suggest these children are more sensitive to these additives and tend to have repetitive eating behavior, which makes them more likely to opt for processed foods, influenced by the vibrant colors absent in natural foods. Given that childhood is a crucial phase of ongoing development, this food selectivity can result in growth delays, malnutrition, degenerative problems, dependence on dietary supplements and difficulties in social interactions, putting them at greater risk compared to non-autistic children [5].

Studies with specific population groups highlight that the antagonistic activity of synthetic additives can be considered a public health problem, which requires increased attention from public authorities. This concern is attributed to research results by authors such as Basu et al. [21], who discovered how the dye known as Patent Blue V was able to alter the secondary structure of human hemoglobin. Furthermore, in other studies, numerous activities that are not beneficial to human health associated with exposure to color additives have been reported, including cytotoxic and clastogenic activities and cytostatic potential [14,16].

**Table 1 ijerph-21-00682-t001:** List of the main problems caused in children by the toxicity of artificial colors.

Number	Diagnosis	Main Results	Study
1	ASD	The association between food dyes and hyperactive behavior in children, as well as the correlation between yellow dye and sleep disorders. Zinc (Zn) deficiency. Mercury accumulation and increased oxidative stress.	[22]
2	ADHD	Results obtained: Discovery of the mechanisms and molecular targets behind the neurodevelopmental processes linked to Attention Deficit Hyperactivity Disorder (ADHD), in addition to other symptoms linked to neurological development (such as cognitive function, learning and memory disorders, among others). Decreased synaptogenesis, decreased neuronal network function, neuroinflammation, and neurodegeneration.	[23]
3	Without behavioral disorders and with ADHD	Studies support the relationship between exposure to food dyes and adverse behavioral outcomes in children. Additionally, the animal toxicology literature provides additional support for behavioral effects.	[7]
4	ADHD	Possible mechanisms include artificial food additives that can cause nutritional deficiencies, trigger allergic reactions or interfere with neurotransmitter levels.	[8]
5	No behavioral disorders	The results and the literature corroborate the rarity of hypersensitivity to food additives when this diagnosis is suspected by parents. Of 23 patients with suspected food allergies, only 1 had a formally confirmed allergy to the mixture of dyes E120 and E124.	[24]
6	ASD, ADHD	The literature indicates that exposure to heavy metals and poor diet are crucial epigenetic factors behind the epidemics of autism and Attention Deficit Hyperactivity Disorder in the US.	[25]
7	No behavioral disorders	According to the findings of this study, prolonged use of foods containing sodium benzoate, aspartame, tartrazine, carrageenan and potassium benzoate resulted in teratogenicity and the emergence of other allergens.	[26]
8	ADHD	A growing body of evidence suggests that the behavioral symptoms of subgroups of children with ADHD may improve with the elimination of certain foods.	[27]
9	No behavioral disorders	The finding is that food additives have a greater toxic potential in children due to their lower body weight compared to adults, as evidenced by the available literature.	[28]
10	No behavioral disorders	In the population of school-age children in Saudi Arabia, the artificial food colorings Brilliant Blue (E133) and Tartrazine (E102) were the most consumed. The high intake of these additives may be related to nutritional risk and associated with food safety.	[29]
11	No behavioral disorders	A substantial proportion of parents reported observing child hyperactivity after consuming certain types of foods and drinks, and although they were aware of the presence of colorings in foods regularly consumed by their children, they did not demonstrate great awareness about the long-term harmful effects of these additives.	[30]
12	No behavioral disorders	Additives most commonly found in foods consumed by children include bisphenols, phthalates, perfluoroalkyl chemicals, perchlorates, pesticides, nitrates, nitrites, artificial food coloring, monosodium glutamate, and aspartame.	[31]
13	No behavioral disorders	The intake of FD&C Red No. 40 is twice the US FDA Acceptable Daily Intake (ADI) in some children’s analgesic syrups and nearly three times the US FDA ADI in some cough, cold and allergy syrups.	[32]
14	No behavioral disorders	The presence of FD&C Red No. 40 in a children’s cold, cough and allergy syrup results in twice the usual daily exposure to this food additive for children ages 12 to 16.	[5]
15	No behavioral disorders	Considerable intra- and inter-batch variability across all the FD&C dyes. Red No. 40 was the most prevalent colorant identified across all the food categories. Azo dyes contribute 10–22% of the maximum Acceptable Daily Intake (ADI) in beverages, while FD&C Red No. 40 contributes 7–9% of the maximum ADI in beverages.	[33]

Source: Prepared by the authors (2024).

## 4. Discussion

Most synthetic dyes permitted for use in foods in Brazil have similar chemical structures, based on functional groups characterized by azo, indigoid, triphenylmethane, and xanthine. Six of the eleven dyes used in Brazil belong to the azo class, namely Ponceau 4R, Red 40, Azorubin, Tartrazine and Twilight Yellow. These compounds have the azo group bond (-N=N-) and at least one sulfonated group (SO_3_^−^) to increase solubility in water. These dyes have an acidic character, and their properties are very similar to each other. Chemically, they are easily reducible by strong reducing agents and are therefore not very sensitive to oxidizing agents. As Tartrazine and Carmoisine are nitrous derivatives (azo class), they can be reduced in the body to an aromatic amine that is highly sensitizing [34].

The main metabolite identified to date is sulfanilic acid. Studies have indicated that these dyes may have several toxic properties for the body. Azo dyes such as Tartrazine and Carmoisine can adversely affect and alter biochemical markers in vital organs, for example, the liver and kidney, not only at higher doses but also at low doses. Tartrazine and Carmoisine not only cause changes in liver and kidney parameters, but their effect also becomes riskier at higher doses, because they can induce oxidative stress through the formation of free radicals, radicals that can cause irreversible damage to the DNA of healthy cells, leading to atypical neurodevelopment in children [34].

The origin of Attention Deficit Hyperactivity Disorder has not yet been completely clarified and, to date, genetic predisposition has been identified as one of the main triggering factors [35]. However, studies indicate that environmental elements, such as daily exposure to food additives, can influence the manifestation of symptoms of this disorder [8]. In this context, research conducted by Chappell et al. [23] investigated the potential impact of seven synthetic dyes approved by the United States Food and Drug Administration (FDA), which are used in food coloring, on the symptoms of ADHD. The results revealed that Erythrosine (Red No. 3), one of the dyes released by the FDA, showed evidence of interference with children’s neurodevelopment.

According to Chappell et al. [23], this dye directly amplifies ADHD symptoms and encompasses other related symptoms, such as impaired cognitive function, learning disorders, and memory problems. The research results indicate that Erythrosine acts antagonistically on neurodevelopmental pathways, causing a specific loss in neurotransmitter activity. This change is associated not only with ADHD but also with other behavioral outcomes in children, including characteristic anxiety symptoms. According to the authors, this behavioral change is attributed to several factors associated with the consumption of these industrial dyes, including the hypothesis of dopamine release, nutritional deficiencies and histamine release, all identified as triggers for changes in behavior. Data from the ToxCast/Tox21 high-throughput screening (HTS) program also support these issues related to the food toxicology of the synthetic dye Erythrosine.

The study conducted by Bakthavachalu et al. [22] aimed to understand how artificial food colors can alter zinc (Zn) metabolism in children diagnosed with Autism Spectrum Disorder. The researchers’ concern regarding artificial colorings is due to the ability of these food additives to compromise the functions of Zn in the human body. According to the authors, zinc plays a crucial role in the formation of the nucleic acids responsible for maintaining health; however, they emphasize that children with ASD often have nutritional deficiencies in this element and that a lack of zinc is associated with several pathological conditions, including delayed growth, compromised immunity, as well as neurodegenerative diseases and neurodevelopmental disorders.

As highlighted by Bakthavachalu et al. [22], the lack of zinc, when associated with exposure to mercury, influences the behavior of children with autism due to an increase in oxidative stress. The researchers explain that the decrease in zinc levels in the body in the ASD group occurs due to the presence of neurotoxic chemicals often found in artificial dyes, such as petroleum, formaldehyde, aniline, hydroxides and sulfuric acids. These components found in artificial additives end up hindering the elimination of heavy metals. According to the authors and the literature data, for this process to occur efficiently, the expression of the metallothionein (MT) gene responsible for the synthesis of the metal-binding protein metallothionein, which is a zinc-dependent function, is essential.

A meta-analysis was conducted by Miller et al. [8] in the State of California under the auspices of the California Office of Environmental Health Hazard Assessment (OEHHA). The primary objective of this study was to elucidate the influence of food dyes on children, both those with behavioral disorders and those without. To achieve this objective, the research incorporated a total of 27 studies from the literature, also covering experiments on animals that focused on the toxic effects of synthetic dyes. The results reveal that 64% of the analyzed studies indicate the presence of neurobehavioral changes in the studied children. Among these results, 52% highlight statistically significant effects. It is crucial to highlight that such behavioral effects were not only observed by children’s caregivers but were also identified by teachers who interact daily with students.

Miller et al. [8] also observed that the consumption of dyes, such as Yellow No. 5 (Tartrazine), in the amount of 50 mg/day can induce behavioral changes in children. In addition to this finding, the research revealed that approximately 8% of children with Attention Deficit Hyperactivity Disorder experience an increase in behavioral symptoms when exposed to synthetic dyes. Other research also supports this issue. Additional results coming from the work of Dos Santos et al. [36] corroborate these findings, identifying a series of adverse health effects associated with Tartrazine consumption. These include cytotoxicity in fibroblasts and human gastric cells, together with evidence of the mutagenic effects of food dyes in eukaryotic cells.

Furthermore, animal studies show that rodents exposed to synthetic dyes at different stages of life, from prenatal to adulthood, showed behavioral changes that support the clinical trials carried out in humans in this meta-analysis [8]. This coherence between the results obtained in animal models and clinical studies highlights the robustness of the conclusions, reinforcing the importance of the potential impacts of food dyes on behavioral health. The findings of this study have significant implications for the health and well-being of children, as well as for the formulation of food coloring regulation policies. There is, evidently, a need for a more cautious approach to the use of these additives, especially in products intended for children’s consumption. Furthermore, the results highlight the importance of greater investment in the production of natural dyes, aiming to develop preventive and interventionist strategies.

Excursion to restrictive diets is a common practice in clinical studies when it is suspected that a certain food may trigger or worsen psychic behaviors, gastric discomfort, allergies or other symptoms. In this context, the restrictive diet operates by temporarily excluding the food in question from the participants’ diet, and any changes in behavior after this exclusion are carefully recorded and analyzed. Rambler et al. [2] performed research in this format that sought to associate an increase or decrease in hyperactivity in children diagnosed with ADHD after the consumption and exclusion of artificial dyes Blue No. 1 and Blue No. 2, and they found that the consumption of these additives can increase hyperactivity in children with ADHD.

According to Rambler et al. [2], one-third (33%) of children diagnosed with ADHD could potentially benefit from dietary interventions where synthetic dyes are excluded from the diet. The researchers report that in addition to studies in children, mice and rats were exposed to an amount of Blue No. 1 and Blue No. 2 recommended by the United States Food and Drug Administration, and even so, these animals showed hyperactive behaviors after consuming the Blue No. 1. The researchers’ hypothesis for this mechanism of action is that artificial food colorings can induce nutritional deficiencies, which could result in impaired neuronal development. Researchers such as Ke et al. [37] associate it with the accumulation of heavy metals such as mercury in the bodies of these children due to the ability of synthetic dyes to modify the levels of zinc and manganese.

In a data analysis covering the period from 2006 to 2021, Dufault et al. [25] found a significant increase in the number of children in the United States who required tutoring in specialized education due to learning difficulties. There was a 242% increase in cases of autism and 184% in developmental delays, as well as other health problems. These data raise doubts among researchers about whether the diet of this population was impacting epigenetic inheritance. One of the central concerns refers to synthetic dyes, which the authors attribute to the individual manufacturing processes of these additives, due to the results presented in 2022 by the United States Food and Drug Administration. According to the agency, some of the dyes are derived from petroleum and have low levels of elements such as mercury, arsenic, lead and cadmium in their formulations [38]. However, even though the controller ensures that these levels are safe, the authors took into account research such as that by Dufault [39], which suggests that many of these dyes are exempt from certification, which implies the absence of control over the levels of these elements. It is important to highlight that these components, combined with the presence of Tartrazine (Yellow 5), demonstrate the ability to interfere with zinc absorption in children [40].

As discussed previously, zinc plays a crucial role in the production of metallothionein (MT). These cytosolic proteins have the ability to capture and, consequently, prevent the dissemination of metals throughout the body, thus preventing the toxicological effects associated with them. Based on this understanding, Dufault et al. [25] suggest that deficiency or alteration in the function of metallothionein may be one of the factors contributing to the increase in disorders such as Autism Spectrum Disorder observed in children in the United States population. Understanding the relationship between zinc availability, metallothionein production and the presence of synthetic dyes in the diet is crucial to better understanding and addressing this health problem related to learning and the high demand for specialized education.

The restrictive dietary regime known as the oligoantigenic diet was studied by Lange et al. [27]. In their conclusion, the authors highlight the beneficial effects of this approach in the management of ADHD. The results are considered convincing, as several studies have indicated that this strategy can significantly reduce ADHD symptoms in children. A meta-analysis included in the study by Lange et al. [27] revealed that artificial colors and preservatives trigger symptoms of hyperactivity, in addition to other related symptoms, in approximately 79% of the children analyzed. In this study, a total of 302 children were tested and divided into 2 groups: 153 children aged 3 years old and 149 children aged between 8 and 9 years old. The study, conducted in a double-blind and placebo-controlled manner, followed the following protocol: over the course of 1 day, children consumed mixed fruit juice in quantities of 300 mL/day for 3-year-old children (mixture A) and 625 mL/day for children aged 8/9 years (mixture B).

The mixtures differed both in the quantity and in the addition of additives. The specific measures for each group followed the average permitted daily consumption of these additives by children in these age groups in the United Kingdom. The drinks were enriched with artificial food colors, including Sunset Yellow [E110], Carmoisine [E122], Tartrazine [E102], Ponceau [4R] and sodium benzoate. Consequently, as described by Lange et al. [27], the authors of this study were able to observe that 79% of children experienced behavioral changes, showing an increase in hyperactivity after consuming at least 85% of the drink. The results, as reported by the authors, were based on the z-scores and the sum of the behavioral changes reported in parent and teacher assessments, and they included a computerized observational test for the 8- and 9-year-old group.

The study conducted by Lange et al. [27] encompasses a variety of meta-analyses, each with distinct approaches, but converging on the conclusion that the elimination of artificial colorings proves to be a beneficial strategy in treating the symptoms of Attention Deficit Hyperactivity Disorder, providing improvements in general. Continuing this theoretical path raised by the researchers, another point observed that reinforces the concern related to color additives is, precisely during the reintroduction of foods phase into the oligoantigenic diet, an integral part of the observational process of children’s behavior that occurs immediately after the restriction period, a notable increase in patients’ hyperactivity was identified, especially when reintroducing foods containing color additives. This finding suggests that the presence of these additives in the diet can trigger hyperactive behaviors, highlighting the importance of raising awareness and considering the influence of artificial colors in the context of ADHD.

Concern about the effects of synthetic dyes has led researchers around the world to try to understand the extent to which children can be harmed by daily exposure to these additives. Case reports linking these dyes to allergies and other health problems, such as asthma, Attention Deficit Hyperactivity Disorder, heart problems, cancer and obesity, are being associated with the consumption of synthetic additives [41]. An example is the study carried out by Sambu et al. [26], who highlight the teratogenic effect of Tartrazine, one of the most common food colorings in the food industry, during the critical period of fetal development in rats, which ranges from the sixth to the fifteenth day of gestation. These pregnant rats were exposed to this dye, resulting in a range of symptoms in the fetuses, including fetal resorptions, mortality, cardiomegaly, hepatorenal damage, and splenic pigmentation. Furthermore, skeletal malformations were observed, such as the absence of coccygeal vertebrae, sternebrae and hind limbs, as well as unequal ribs, as a consequence of the treatment.

Kraemer et al. [28] highlight that the toxic effects of synthetic additives are especially worrying in children due to their body proportions, which make them more susceptible to side effects. The authors also emphasize the importance of updated studies on artificial colors approved by the food regulatory agencies in each country. This is due to the fact that results considered outdated for these additives may be used as parameters to keep them considered safe for food consumption. In this sense, in the discussion, the authors cite the issue related to a widely debated topic surrounding the dye titanium dioxide, which was previously approved and considered safe by regulatory agencies but is now banned in some countries due to its toxic effects on human health. Since 2021, this dye has been the subject of studies on its safety, as mentioned by Younes et al. [42]. In Brazil, the ANVISA has also carried out investigations into the consumption of titanium dioxide [43].

A study carried out in Saudi Arabia involved 5000 children between the ages of 6 and 17 and revealed a high consumption of products containing synthetic food colorings, including some not permitted in the country [29]. Initially, the researchers used 24 h food frequency questionnaires to identify the products most consumed by the group. A total of 839 products were noted, such as cookies, cakes, chocolates, snacks, ice cream, juices and drinks, sweets, jellies and chewing gum. The results revealed a greater consumption of juices and drinks, ice cream and cakes among the foods studied. During the analysis, the researchers observed the presence of ten different types of synthetic dyes, two of which were not allowed: Erythrosine (E127) and Red 2G (E128). The collected foods were subjected to high-performance liquid chromatography together with a diode array detector.

Mohammed et al. [29] also observed that the food colorings most consumed by this population were Brilliant Blue (E133) and Tartrazine (E102), with significant consumption percentages of 54.1% and 42.3%, respectively. Furthermore, high percentages of Sunset Yellow (E110) outside the permitted limits were detected in chocolates, as well as Tartrazine in two drinks. The authors expressed concern about the high consumption of ultra-processed foods in this population due to the associated nutritional risks, which are directly linked to food safety. The authors’ narrative is similar to other researchers’ concerns in different countries, because studies indicate the presence of petroleum-based ingredients in the formulation of these products [44]. Another point raised in the research to date is the lack of understanding about the potential risks associated with mixing different types of synthetic dyes. Furthermore, there are no studies that clearly demonstrate the possible long-term side effects of consuming these dyes.

Even children without behavioral disorders can suffer some type of change when they consume synthetic food additives daily, says Miller et al. [8]. However, few caregivers know this information. A group of parents, after undergoing a study on food containing synthetic dyes in the city of Jazan, Saudi Arabia, observed altered behavior in their children immediately after consuming foods and drinks containing these substances. A total of 88.9% reported hyperactivity, in addition to associating the condition with specific food groups such as soft drinks (35.61%), chocolates (31.82%) and sweets and candies (30.3%) [30]. The authors’ objective was to investigate the perception of this population in relation to the dyes present in foods and drinks, and the relationship with the behavior of their children. In the study, 387 people were interviewed, with more than 75% of the group being young and more than 60% having a higher level of education. Surprisingly, only a small percentage, less than 3% of participants, reported being aware of the harms of color additives for children.

Returning to what was previously discussed, in the study conducted by Makeen et al. [30], titled “Parents’ perception of the role of colored food additives on children’s behavior in the Jazan region of Saudi Arabia”, the authors presented significant data on the relationship between food additives and hyperactivity in children. However, they did not address the possible relationships with the sugars present in these foods. It is important to note that food groups such as soft drinks, candies and chocolates are often associated with hyperactive behaviors, as they contain high levels of sugar. Researchers like Johnson et al. [45] and Del-Ponte et al. [46] have already demonstrated correlations between sugar consumption and symptoms related to ADHD. Furthermore, the analysis carried out by Paglia et al. [47] suggests that physiologically, in children, sugar is rapidly absorbed into the bloodstream, resulting in rapid fluctuations in glucose levels and initiating the production of adrenaline.

The aforementioned findings reveal a significant gap in the knowledge of parents, caregivers and the general population about the potential adverse effects of synthetic dyes found in food. As new research is completed and unprecedented discoveries about the possible harm to health related to these color additives are presented, the view of many researchers that this topic needs to be more widely debated and clarified to consumers in general is reinforced. Furthermore, following research models, such as the study carried out by Mohammed et al. [29], which used high-performance liquid chromatography, can reveal the unsuitability of certain foods produced by the food industry. This type of approach can be crucial for identifying and remedying problems in food composition and safety, thus contributing to the promotion of public health.

The findings of Savin et al. [31] on the effects of synthetic dyes included symptoms of hypersensitivity, behavioral disturbances, and attention deficits in children, as well as a decline in learning performance. An intriguing aspect highlighted in this research is the variability in children’s sensitivity to food dyes, with some being more sensitive than others. The finding is based on the conclusions of Miller et al. [8]. Researchers have identified that children with certain genetic variations in genes involved in metabolizing histamine have more severe reactions to synthetic color additives. This is because histamine works as a neurotransmitter, playing a vital role in alertness and in triggering hypersensitivity reactions.

Due to these concerns, Lehmkuhler et al. [32], Thilakaratne et al. [33] and Lehmkuhler et al. [48] converge on a common concern: the presence and levels of synthetic food coloring in over-the-counter foods, medications, and vitamins intended for pediatrics and pregnant women in the United States, in addition to what is permitted by the FDA, Food Drug and Cosmetic (FD&C), and the World Health Organization (WHO). All the authors used high-performance liquid chromatography with a diode array photometric detector (HPLC-PDA) to investigate the levels of these additives in products that included infant and prenatal vitamins in the form of gums and tablets, cough, colds and allergies syrups, and painkiller tablets. Lehmkuhler et al. [48], in particular, also examined coloring levels in 10 food categories that included juices, cereals, desserts, ice cream cones, soft drinks, toppings, salty snacks, and fruit jams, among others.

In 2020, Lehmkuhler et al. [32] identified a significant gap in the research, noting the absence of studies evaluating the dye levels in medications and vitamin supplements in the United States. This is especially worrying, considering that the color additives used for this purpose are the same as those used in foods, and several previous studies have already suggested a possible association between the consumption of these dyes and changes in the neurobehavioral behavior of children. Another factor that drove this investigation was the lack of transparency regarding the amount of dyes used in pharmaceuticals, which is often protected by patents. This made it challenging to estimate children’s exposure to dyes through dietary intake. Given this scenario, the researchers decided to evaluate the levels of FD&C Red No. 40, Yellow No. 5, Yellow No. 6, Blue No. 1 and Blue No. 2 dyes.

According to the evidence presented by Lehmkuhler et al. [32], there are considerable variations in the additive levels between different manufacturers. Notably, the highest concentrations of dyes were found in painkillers and cough, cold, and allergy syrups. This high concentration in syrups can be attributed to the strategy of making products more attractive to children, using vibrant colors to encourage consumption. However, it is concerning to note that, especially in syrups, the acceptable daily intake of FD&C Red No. 40 dye is almost three times the limit established by the US FDA. Furthermore, in vitamins in the form of chewing gum, the intake of Red No. 40, Blue No. 1 and Yellow No. 6 dyes is about twice as high as allowed.

After the study carried out in 2020 on medicines, Lehmkuhler et al. [48] conducted an analysis in 2023, using HPLC, of the levels of synthetic dyes in foods consumed predominantly by children and young people in the ten categories previously identified. Once again, FD&C Red No. 40 was widely detected. According to the authors, all the foods analyzed contained this dye, but its presence did not exceed the limits established by the FDA.

However, it was observed that FD&C Red No. 40 represented between 7% and 9% of the maximum Acceptable Daily Intake (ADI) in the United States, specifically in the beverage categories per serving. For context, recommendations suggest that a child between 6 and 10 years of age should not exceed 10% of the ADI in each serving throughout the day. Therefore, the presence of FD&C Red No. 40 at levels between 7% and 9% of the maximum ADI value in drinks per serving suggests an exposure close to or even above the recommended limit for this age group.

The study carried out by Lehmkuhler et al. [46] appears to corroborate the findings of Thilakaratne et al. [47], despite being carried out three years apart. Both studies adopted similar methodologies in analyzing over-the-counter medications intended for children and pregnant women in the United States. Both studies found concerning levels of FD&C Red No. 40 in the cough, cold and allergy syrup class, which exceed the limits considered safe by FD&C. Thilakaratne et al. state that, if a child between 12 and 16 years of age consumed these medications in the doses recommended by the leaflet, they would be ingesting 0.221 mg/kg/day of the dye, which represents twice the recommended exposure for this age group. These findings highlight the importance of greater surveillance and regulation of dyes used in medicines intended for sensitive groups, such as children and pregnant women, in order to ensure safety and minimize health risks.

The choice of the high-performance liquid chromatography technique allows for a precise and sensitive analysis of dyes, guaranteeing reliable results to support regulatory decisions and clinical practices. The finding that these dyes often exceed the limits considered safe by the FDA raises serious concerns about potential health impacts, especially in children. Excessive exposure to these dyes may not only be associated with changes in neurobehavioral behavior, as suggested by previous studies, but may also pose long-term health risks. Therefore, these research findings highlight the urgent need for stricter regulation and greater transparency in the food and pharmaceutical industry, aiming to protect public health and ensure that consumers, especially the most vulnerable, are not exposed to unnecessary risks.

When comparing the results obtained in this study with other research in the same line of investigation, a significant connection is noted between the intake of food coloring and adverse behaviors in children. This article also reinforces literary data that highlight the influence of epigenetic factors, such as exposure to heavy metals and unbalanced diets, on the etiology of neurodevelopmental disorders such as autism and ADHD, specifically highlighting the accumulation of mercury and its association with oxidative stress and neurodegenerative damage. The main insights arising from the comparison between studies concern the association of two or more food additives, such as sodium benzoate and tartrazine, which together are associated with teratogenicity and the development of allergies, although in some studies the effects are attributed only to tartrazine.

At the same time, we observed a gap in most studies that failed to mention the effects of the sugar content present in most pigmented foods used mainly in double-blind studies. The authors barely mentioned the contribution of this additive to the increase in hyperactivity, as indicated by studies by Johnson et al. [42] and Del-Ponte et al. [43]. Another point to be highlighted is regarding the presence of dyes in children’s medicines. We found three studies that analyzed the content of dyes in medications, but we found reviews in the literature raising concerns about excessive exposure in pharmaceuticals, highlighting the importance of stricter regulations and continued surveillance to ensure food safety and children’s health.

The limitations of studies regarding the effects of synthetic dyes can be varied and comprehensive. One of them is the lack of comprehensive and specific research on the long-term effects of these dyes in different populations, especially in children and individuals with food sensitivities. Additionally, the diversity of synthetic dyes used in the food industry hampers the generalization of results. The lack of standardization in the methods used to assess the effects can also compromise the validity and comparability of studies. Another common limitation is the difficulty of isolating the effects of synthetic dyes from other food components, making the attribution of causes and effects complex. Furthermore, the influence of individual factors, such as diet and lifestyle, is often not fully controlled in studies, which can affect the interpretation of results. These limitations highlight the need for a more comprehensive and rigorous approach when investigating the effects of synthetic dyes on human health.

Given this scenario, there has been growing interest in replacing synthetic dyes with more natural and sustainable options. A promising alternative is the use of natural dyes, derived from plant and animal sources. These dyes are generally considered safe for human consumption, in addition to having antioxidant and nutritional properties [1]. Another growing category comprises organic dyes, which are obtained from organic compounds and offer a more eco-friendly option, often produced using biotechnological methods. From this perspective, Chatragadda et al. [49] has explored the potential of microbial dyes, produced by bacteria, as a viable alternative to synthetic dyes. These emerging options have the potential to not only reduce the environmental impacts associated with dye production but also offer a healthier and more sustainable choice for consumers concerned about their health and the environment.

## 5. Conclusions

The harms of synthetic dyes in children, both with and without comorbidities, are worrying and require a careful and proactive approach. The studies presented in this review demonstrate an association between the consumption of these additives and a variety of adverse health effects, ranging from hyperactivity and behavioral disorders to allergic reactions and more serious health problems. For children with comorbidities such as Autism Spectrum Disorder and Attention Deficit Hyperactivity Disorder, exposure to synthetic dyes can worsen existing symptoms and complicate management of the condition. Furthermore, even in children without comorbidities, synthetic dyes can pose a health risk by affecting their cognitive, behavioral, metabolic and nutritional development. Given these facts, educational approaches for parents, caregivers and health professionals are essential so that they are aware of the potential risks associated with the consumption of synthetic dyes and adopt measures to reduce children’s exposure to these additives. Raising awareness through campaigns and better food labeling informing about the types of dyes present in each specific product, as well as the formulation of stricter policies to control the recommended quantity, are ways of mitigating the possible toxicity levels of these additives. Another way is to promote a more natural and balanced diet and search for dyes from completely natural sources.

## Figures and Tables

**Figure 1 ijerph-21-00682-f001:**
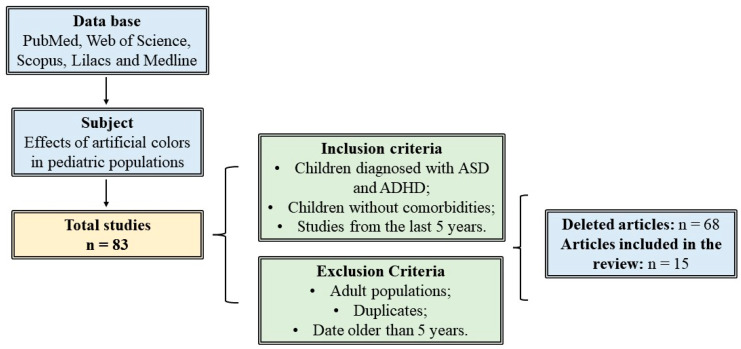
Flowchart representing the stages of article selection and inclusion.

## Data Availability

The data associated with this study have not been deposited in a publicly available repository; however, the data can be provided upon request.

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
