# Peer review of "Synthetic Colors in Food: A Warning for Children’s Health"

_ijerph, 2024, doi:10.3390/ijerph21060682_

Round 1

Reviewer 1 Report

Comments and Suggestions for Authors

The paper's topic is relevant and the perspective could be valuable for those interested. However, the manuscript needs to improve—a few comments. A deeper understanding of the molecular characteristics of synthetic dyes and the overall mechanistic way in which the compounds bind to cellular receptors or what other kinds of chemical reactions within the body could be of great value. Also, I recommend including and improving the reach that other reviews on the same topic have worked on previously; other authors' points of view could enhance the scientific merit of this work. The manuscript merely describes the relevant works on the materials. However, it lacks a critical analysis of the analyzed references since it has yet to show a new perspective on applying the current knowledge in technological applications. In a general way, the reviews must criticize the previous work and provide new insights into the topics, which in this case is restricted to 1 paragraph. In the way it is written, it does not clearly show the advances and their application or possible use of such information, and it seems like it's trying to deliver a message, but it is not clear both in intention and delivery.

Author Response

Manuscript ID: IJERPH-2931684

Submission Title: Synthetic Colors in Food: A Warning for Children's Health

Dear Reviewer,

Thank you very much for your consideration of our manuscript. We appreciate the thoughtful comments and suggestions offered by you.  We have incorporated a number of suggestions into the manuscript. We have highlighted (yellow-marked) all changes in the manuscript to help to identify corrected text in the manuscript.  Please also find below a point-by-point response to the comments.

On behalf of my coauthors, I thank you for your time and look forward to your response.

Best regards,

Zandleme Birino de Oliveira.

Response to Reviewer

In response to Reviewer 1’s comments, please find attached the revised version for your consideration.

The paper's topic is relevant and the perspective could be valuable for those interested. However, the manuscript needs to improve a few comments.

Response: Thank you for your comments. We agree that some points need to be improved or changed.

A deeper understanding of the molecular characteristics of synthetic dyes and the overall mechanistic way in which the compounds bind to cellular receptors or what other kinds of chemical reactions within the body could be of great value.

Response: Thanks for your comment. We have added two paragraphs with a better understanding of the molecular aspects of synthetic dyes and how they can cause damage to cells. We believe that the added information is of great value for a better understanding of the mechanism of action of dyes at a cellular level. The added paragraphs can be checked on the lines 159 to 176.

Also, I recommend including and improving the reach that other reviews on the same topic have worked on previously; other authors' points of view could enhance the scientific merit of this work. The manuscript merely describes the relevant works on the materials. However, it lacks a critical analysis of the analyzed references since it has yet to show a new perspective on applying the current knowledge in technological applications. In a general way, the reviews must criticize the previous work and provide new insights into the topics, which in this case is restricted to 1 paragraph. In the way it is written, it does not clearly show the advances and their application or possible use of such information, and it seems like it's trying to deliver a message, but it is not clear both in intention and delivery.

Response: We appreciate your observations. In the new version of the manuscript, we made a series of changes that aim to improve the discussion of the presented and reviewed works. We carried out a critical analysis of the references analyzed, which can be seen in table 1 (marked in yellow). We have improved the discussion (marked in yellow), so that the advances and application of the information obtained in our study are clearly defined. Furthermore, we created a new paragraph about the main message that our work intends to convey to readers (lines 482 to 492).

Reviewer 2 Report

Comments and Suggestions for Authors

Zandleme Birino de Oliveira and co-worker presented an interesting data on Synthetic Colors in Food: A Warning for Children's Health, I found that paper is well drafted however, some concerns are their that need to address before moving for next process.

Suggested to draw a flow chart indicating how the systematic analysis was performed using key words/bullion term, exclusion criteria, etc.

Line no 130-131: please check what is this "contained in the second panel. Figures should be placed in the main text near to the first time they are cited". and need to delete.

Line no. 133-134 "Source: Oliveira ZB, et al., 2024" is table has been taken from this source, then needs to check whether permission require or not.

Suggested to elaborate this children's neurodevelopment, Autism Spectrum Disorder and Avoidant/Restrictive Food Intake Disorder in introduction section, so reader may get more attention on this sensitive topic which is going to affect directly future.

Table 1: suggested to remove title of the study and elaborate further critical attributes presented in that paper in result section of the table 1.

Suggested to addon on alternative to synthetic dye as food colorant before conclusion section

Thanks 

Author Response

Manuscript ID: IJERPH-2931684

Submission Title: Synthetic Colors in Food: A Warning for Children's Health

Dear Reviewer,

Thank you very much for your consideration of our manuscript. We appreciate the thoughtful comments and suggestions offered by you.  We have incorporated a number of suggestions into the manuscript. We have highlighted (yellow-marked) all changes in the manuscript to help to identify corrected text in the manuscript.  Please also find below a point-by-point response to the comments.

On behalf of my coauthors, I thank you for your time and look forward to your response.

Best regards,

Zandleme Birino de Oliveira.

Response to Reviewer

In response to Reviewer 2’s comments, please find attached the revised version for your consideration.

Zandleme Birino de Oliveira and co-worker presented an interesting data on Synthetic Colors in Food: A Warning for Children's Health, I found that paper is well drafted however, some concerns are their that need to address before moving for next process.

Response: Thank you for your comments. We agree that some points need to be improved or changed.

Suggested to draw a flow chart indicating how the systematic analysis was performed using key words/bullion term, exclusion criteria, etc.

Response: Thanks for your suggestion. We added a flowchart detailing the methodology and inclusion and exclusion criteria in the interactive bibliographic review that we carried out in our study. The flowchart can be seen in the material and methods section, line 84.

Line no 130-131: please check what is this "contained in the second panel. Figures should be placed in the main text near to the first time they are cited". and need to delete.

Response: Thank you for your observation. We checked the line in question and realized that it was a formatting error. We deleted the section.

Line no. 133-134 "Source: Oliveira ZB, et al., 2024" is table has been taken from this source, then needs to check whether permission require or not.

Response: Thanks for the observation. In fact, the source in question is the authors of the manuscript themselves, since the table was prepared by the authors. However, we changed the presentation of the table source, marked in yellow, line 156.

Suggested to elaborate this children's neurodevelopment, Autism Spectrum Disorder and Avoidant/Restrictive Food Intake Disorder in introduction section, so reader may get more attention on this sensitive topic which is going to affect directly future.

Response: Thank you for your comment and suggestion. In the introduction, we added new information about the neurodevelopment of children with ASD and ADHD, marked in yellow, lines 40 to 56.

Table 1: suggested to remove title of the study and elaborate further critical attributes presented in that paper in result section of the table 1.

Response: Thanks for your suggestion. As suggested, we removed the title of the work presented in table 1 (marked in yellow), and added new critical attributes about each work described in table 1. Check the changes in column 3 of table 1, marked in yellow.

Suggested to addon on alternative to synthetic dye as food colorant before conclusion section.

Response: Thanks for your suggestion. As suggested, we added a paragraph before the conclusion, presenting alternatives to the use of synthetic dyes in foods (marked in yellow, lines 482 to 492).

Reviewer 3 Report

Comments and Suggestions for Authors

I consider that this manuscript could be accepted for publication in IJERPH, although previously authors should clarify and deal with some aspects in relation to it.

-          Line 52. The reference is not appropriately cited. Please revise how references are cited in other manuscripts also published in this journal or even instructions for authors. The same for references cited in lines 61, 104, 125, 141, 146, 158, 167, 186, 210, 215, 222, 256, 273, 281, 291, 303, 316… Please revise all cites along the manuscript.

-          Line 63. Explain in a detailed way how do you check the methodological quality of the articles included in this review.

-          Line 70. If for different descriptors authors use capital letter for the first letter of the word the same should be applied for “Toxicity”.

-          In my opinion the number of articles (only 11) selected for this review is small.

-          Lines 130-131. This sentence is inadequately placed.

-          Line 156. Do not repeat “Red No. 3”. It has previously referred.

-          Lines 231-235. Did authors of the referred reference [25] measured the levels of reported elements namely Hg, As, Pb and Cd due to that synthetic dyes are derived form petroleum? In negative case, how do authors think that these other researchers  arrive to this conclusion?

-          Lines 236-238. The MT is not directly responsible for eliminating heavy metals but for the tolerance of the body to them. In this sense the MT has the capacity to sequester and avoid the organism distribution of these heavy metals and therefore the toxicological effects associated to them.

-          Lines 252-255. What type of drink was used in this study? It was a coke drink? In this drink the only one additive used was the synthetic dye or there were other additional ones?

-          Lines 259-261. Portuguese language has been used.

-          Line 297. Please correct “group839”.

-          Lines 320-322. Authors reported that parents referred hyperactivity for their children after consuming “soft drinks (35.61%), chocolates (31.82%) and sweets and candies (30.3%)”. Nevertheless it is also known that the consumption of high amounts of sugar present in foods is also related with the reported hyperactivity. What do authors believe in relation to it?

-          Please read guide for authors in relation to the way to cite the used references and unify all them taking into account this information.

Author Response

Manuscript ID: IJERPH-2931684

Submission Title: Synthetic Colors in Food: A Warning for Children's Health

Dear Reviewer,

Thank you very much for your consideration of our manuscript. We appreciate the thoughtful comments and suggestions offered by you.  We have incorporated a number of suggestions into the manuscript. We have highlighted (yellow-marked) all changes in the manuscript to help to identify corrected text in the manuscript.  Please also find below a point-by-point response to the comments.

On behalf of my coauthors, I thank you for your time and look forward to your response.

Best regards,

Zandleme Birino de Oliveira.

Response to Reviewer

In response to Reviewer 3’s comments, please find attached the revised version for your consideration.

I consider that this manuscript could be accepted for publication in IJERPH, although previously authors should clarify and deal with some aspects in relation to it.

Response: Thank you for your comments. We agree that some points need to be improved or changed.

Line 52. The reference is not appropriately cited. Please revise how references are cited in other manuscripts also published in this journal or even instructions for authors. The same for references cited in lines 61, 104, 125, 141, 146, 158, 167, 186, 210, 215, 222, 256, 273, 281, 291, 303, 316… Please revise all cites along the manuscript.

Response: We thank you for your observations. We reviewed all citations in the manuscript and corrected them according to the instructions for authors. All references presented in your comment have been corrected.

Line 63. Explain in a detailed way how do you check the methodological quality of the articles included in this review.

Response: Thanks for your comment. In order to clarify how the methodological quality of the articles was evaluated, we created a flowchart detailing the criteria used for inclusion and exclusion of the articles used in our study. The flowchart can be seen in the methodology section, line 84.

Line 70. If for different descriptors authors use capital letter for the first letter of the word the same should be applied for “Toxicity”.

Response: Thanks for your comment. We corrected the words Toxicity. (marked in yellow, line 81).

In my opinion the number of articles (only 11) selected for this review is small.

Response: Thank you for your comment. We agree with your opinion, however, according to the inclusion criteria established in our methodology, the 11 articles were selected to comply with the publication period of the last 5 years. However, in a new analysis, we were able to add 4 more articles to our study, reaching a total of 15 articles analyzed and discussed in our review. The added articles can be seen at the end of table 1 (numbers 12 to 15).

Lines 130-131. This sentence is inadequately placed.

Response: Thank you for your observation. We checked the line in question and realized that it was a formatting error. We deleted the section.

Line 156. Do not repeat “Red No. 3”. It has previously referred.

Response: Thank you for your observation. We removed the repetition "Red nº. 3" from the sentence. (Line 196).

Lines 231-235. Did authors of the referred reference [25] measured the levels of reported elements namely Hg, As, Pb and Cd due to that synthetic dyes are derived form petroleum? In negative case, how do authors think that these other researchers  arrive to this conclusion?

Response: Thank you for your comment and observation. The authors in question did not measure the levels of the elements, however, the authors took into account research such as that by Dufault, which suggests that many of these dyes are exempt from certification, which implies the absence of control over the levels of these elements. We added this information in the aforementioned paragraph, marked in yellow, lines 270 to 278.

Lines 236-238. The MT is not directly responsible for eliminating heavy metals but for the tolerance of the body to them. In this sense the MT has the capacity to sequester and avoid the organism distribution of these heavy metals and therefore the toxicological effects associated to them.

Response: Thank you for your observation. We have modified the paragraph in question, adding the information you suggested. (Lines 282 to 284).

Lines 252-255. What type of drink was used in this study? It was a coke drink? In this drink the only one additive used was the synthetic dye or there were other additional ones?

Response: Thank you for your comment and observation. The drink in question is a mixed fruit juice. This and other information was added in the aforementioned paragraph, marked in yellow (lines 297 to 302).

Lines 259-261. Portuguese language has been used.

Response: Thank you for your observation. We corrected that sentence. Lines 316 to 318.

Line 297. Please correct “group839”.

Response: Thanks for the observation. We made the suggested correction. Marked in yellow, line 354.

Lines 320-322. Authors reported that parents referred hyperactivity for their children after consuming “soft drinks (35.61%), chocolates (31.82%) and sweets and candies (30.3%)”. Nevertheless it is also known that the consumption of high amounts of sugar present in foods is also related with the reported hyperactivity. What do authors believe in relation to it?

Response: Thank you for your observations. According to some studies, the consumption of foods rich in sugar may also be related to hyperactivity in children. We have added a paragraph discussing this topic in our discussion. Marked in yellow, lines 386 to 397.

Please read guide for authors in relation to the way to cite the used references and unify all them taking into account this information.

Response: Thanks for the suggestion. We read the instructions for authors and standardized all citations in the manuscript.

Round 2

Reviewer 2 Report

Comments and Suggestions for Authors

The authors have reflected all the said suggestions and comments, which made the manuscript enhanced with improved readability; Thus I suggest for further consideration with acceptance.

Reviewer 3 Report

Comments and Suggestions for Authors

Accept.